# Interpreting coronary artery disease GWAS results: A functional genomics approach assessing biological significance

**Katherine Hartmann[1¤]\*, Michał Seweryn[2], Wolfgang Sadee[1]**

**1** Department of Cancer Biology and Genetics, Center for Pharmacogenomics, College of Medicine, The Ohio State University, Columbus, OH, United States of America, **2** Biobank Lab, Department of Molecular Biophysics, University of Lodz, Lodz, Poland

¤ Current address: Department of Radiology, University of Pennsylvania, Philadelphia, PA, United States of America

\* katherine.hartmann@pennmedicine.upenn.edu

**Data Availability Statement:** All CATHGEN and GTEx data files are available from the dbGaP database (accession number(s) phs000703, phs000424.). 1000 genomes data was downloaded

## Abstract

Genome-wide association studies (GWAS) have implicated 58 loci in coronary artery disease (CAD). However, the biological basis for these associations, the relevant genes, and causative variants often remain uncertain. Since the vast majority of GWAS loci reside outside coding regions, most exert regulatory functions. Here we explore the complexity of each of these loci, using tissue specific RNA sequencing data from GTEx to identify genes that exhibit altered expression patterns in the context of GWAS-significant loci, expanding the list of candidate genes from the 75 currently annotated by GWAS to 245, with almost half of these transcripts being non-coding. Tissue specific allelic expression imbalance data, also from GTEx, allows us to uncover GWAS variants that mark functional variation in a locus, *e.g.*, rs7528419 residing in the *SORT1* locus, in liver specifically, and rs72689147 in the *GUYC1A1* locus, across a variety of tissues. We consider the GWAS variant rs1412444 in the LIPA locus in more detail as an example, probing tissue and transcript specific effects of genetic variation in the region. By evaluating linkage disequilibrium (LD) between tissue specific eQTLs, we reveal evidence for multiple functional variants within loci. We identify 3 variants (rs1412444, rs1051338, rs2250781) that when considered together, each improve the ability to account for LIPA gene expression, suggesting multiple interacting factors. These results refine the assignment of 58 GWAS loci to likely causative variants in a handful of cases and for the remainder help to re-prioritize associated genes and RNA isoforms, suggesting that ncRNAs maybe a relevant transcript in almost half of CAD GWAS results. Our findings support a multi-factorial system where a single variant can influence multiple genes and each genes is regulated by multiple variants.

## Introduction

Genome-wide association studies (GWAS) have identified dozens of genetic variants (SNPs) associated with cardiovascular disease risk and related clinical phenotypes (*e.g.*, blood pressure,

from http://ftp.1000genomes.ebi.ac.uk/vol1/ftp/
release/20130502/supporting/GRCh38_positions/.

**Funding:** This study was supported by National
Institutes of Health National Institute of General
Medical Science Pharmacogenetics Research
Network [Grant U01 GM092655 awarded to WS]
https://www.nigms.nih.gov/ and the National
Center for Advancing Translational Sciences [TL1
TR001069 awarded to KH] https://ncats.nih.gov/.

**Competing interests:** The authors have declared
that no competing interests exist.

lipid levels) [1–3]. However, these findings do not necessarily translate to understanding of heritability, likely because we do not fully understand the link between significant loci, causative genetic variants and complex phenotypes [4]. Moreover, the functional variant and even the relevant gene close to a significant locus in many cases remain uncertain. The majority of statistically significant SNPs reside in non-coding regions with poorly defined biological functions and a complex architecture of multiple genes and transcripts [5]. Gene assignment is largely based on proximity, usually with little consideration for non-coding transcripts in the locus or the possibility of chromatin looping that places distant regions in close proximity [6], with regulatory domains often interacting with multiple genomic target regions (9). Additionally, localization to non-coding regions means the mechanisms remain unknown as the function is not immediately obvious, while implicating epigenetics and other regulatory processes [5,7]. This uncertainty limits the utility of GWAS findings. To interpret and refine GWAS results for coronary artery disease (CAD), we use RNA expression, in addition to physical position, to prioritize the variants and gene(s) most likely to be relevant.

Although largely thought of in a single SNP–single protein-coding gene paradigm, GWAS variants mark regions with various degrees of complexity often including several protein-coding and non-coding RNAs (ncRNAs). SNPs located within RNA exons may not only alter the protein sequence but also influence RNA structure and function in a transcript specific manner [8]. Some of these GWAS loci consist of gene clusters that are coordinately regulated [9], and almost all include multiple RNA isoforms expressed from a given gene, including splice isoforms. Within such multi-gene regions, a single variant may affect more than one gene, both protein-coding and non-coding, via chromatin looping between multiple sites or by regulating DNA accessibility for the entire region [9,10]. Therefore, a critical question for interpreting GWAS associations is which gene(s), and what specific transcript(s), are affected within each significant locus.

The potential for multiple variants to affect a single gene is also critical to the interpretation of GWAS. Such interactions between variants, either linear or dynamic (epistasis) and dictated by linkage disequilibrium (LD), may remain hidden in GWAS because of the restrictive nature of multiple hypotheses corrections; however targeted analysis of loci reveals multiple interacting variants modulating gene expression [9,11,12]. Failure to identify all main functional variants in a gene locus and their interactions results in false estimates of the genetic influence of a locus, and further impedes discovery of dynamic interactions that are sensitive to partial or confounded estimates [13–18].

Detailed analysis of RNA expression to evaluate GWAS results is increasingly employed to evaluate co-localization of GWAS and eQTL signals [19–22]. However, most methods rely on the a *priori* assumptions that variants are independent of each other (*e.g.*, eCAVIAR), while COLOC assumes that there is only one functional variant per GWAS locus. These assumptions do not allow for a multifactorial system, where a single variant can influence multiple genes and each gene can be regulated by multiple variants. Accordingly, we search for overlap between variants marking GWAS associations and those marking eQTLs rather than using existing methods to co-localize signals. Although this approach limits our power to detect overlap as it requires a single variant appear as a marker in both GWAS and eQTL analysis, we posit it facilitates functional exploration of a multi-factorial system.

A recent CAD GWAS used 1000 genomes to impute insertions/deletions, rare variants and common variants that were not directly genotyped as part of a large-scale meta-analysis of 185 thousand cases and controls [1]. While confirming 47 of the 48 previously identified loci, this study identified an additional 10 at genome-wide significance, bringing the total count of CAD associated loci to 58. Each of these loci are based on robust statistical associations for one or more SNPs in the locus. Furthermore, each locus has been assigned one or more genes

based largely on proximity as part of the GWAS annotation. We consider each of these 58 loci in detail, using QTL and position to re-prioritize candidate genes and focusing on a subset of loci, to begin resolving inherent complexities of genomic architecture.

## Materials and methods

### Data

Our approach systematically utilizes and combines publicly available information based on the following datasets. Please note each dataset was considered separately; a meta-analysis was not undertaken.

**CARDIoGRAMplusC4D Consortium GWAS results (Nikpay et al).**  GWAS variants, annotated genes, and effect alleles were taken from Supplementary Tables 2 (CAD meta-analysis additive association results for 48 loci previously identified at genome-wide significance) and 4 (Association results of the 10 novel CAD loci including the dominant model) [23]. For these analyses CAD had been defined broadly as those participants with a diagnosis of myocardial infarction, acute coronary syndrome, chronic stable angina or coronary stenosis > 50%.

**1000 genomes.**  Genotypes for calculating LD between SNPs of interest were downloaded from: http://ftp.1000genomes.ebi.ac.uk/vol1/ftp/release/20130502/supporting/GRCh38_positions/. Individuals of the 'EUR' superpopulation were selected for LD calculations.

**CATHeterization GENetics (CATHGEN).**  Expression, genotypes, and clinical phenotypes were acquired via dbGaP Project #5358 (dbGaP accession phs000703). Expression levels had been determined using Illumina HumanHT-12-v3 in RNA from whole blood. We considered variables recorded in pht003672: age (phv00197199), gender (phv00197207), hypercholesterolemia (phv00197204), smoking (phv00197208), number of diseased vessels (phv00197295), CAD Index (phv00197202) and history of myocardial infarction (phv00197212). Approximating the definition of CAD used in the CARDIoGRAMplusC4D Consortium GWAS by Nikpay et al., we defined CAD as history of myocardial infarction and/or vessel occlusion >50% (CAD Index). We restricted analysis to Caucasians (race (phv00197206)) for sample size considerations (862 Caucasians; 259 African Americans). The approach developed here can be extended to other ethnic groups as these datasets become available. Data access was approved by the Ohio State University IRB (Protocol #2013H0096).

**Genotype and Tissue Expression Project (GTEx).**  Tissue-specific RNAseq data was acquired via dbGaP Project #5358 (dbGaP accession phs000424). For details see Lonsdale et al. and http://www.gtexportal.org/home/documentationPage [24]. P-values, effect sizes, and directionality for eQTLs were downloaded directly from the GTEx Portal from the already completed and published analysis of tissue specific eQTLs. Briefly, p-values reflect the alternative hypothesis that the slope of linear regression models accounting for tissue specific normalized gene expression with individual genetic variants is non-zero. This analysis included filters based on overall gene expression, normalized gene expression values, and incorporated covariates including top 5 principal components, covariates identified using Probabilistic Estimation of Expression Residuals (PEER) factors, sequencing platform (Illumina Hiseq 2000 or Hiseq X), sequencing protocol (PCR-based, PCR-free), and gender. CAD was defined as recorded history of heart disease (MHHRTDIS) or heart attack (MHHRTATT) to best approximate the definition used in the CARDIoGRAMplusC4D Consortium GWAS by Nikpay et al. Data access was approved by the Ohio State University IRB (Protocol #2013H0096).

### Gene information

Transcripts, coding status, GO Ids, number of publications indexed in PubMed, gene/transcript expression, GWAS variants, GTEx eQTLs (expression quantitative trait loci) and sQTLs

(splicing quantitative trait loci) including tissue specific expression, and allelic ratios in DNAse hypersensitivity sites were obtained for each gene using the package 'mglR' implemented in R (https://cran.r-project.org/web/packages/mglR/index.html). Protein-coding transcripts were defined as those annotated by BiomaRt as "IGC gene", "IGD gene", "IG gene", "IGJ gene", "IGLV gene", "IGM gene", "IGV gene", "IGZ gene", "nonsense_mediated_decay", "nontranslating CDS", "non stop decay", "polymorphic pseudogene", "TRC gene", "TRD gene", "TRJ gene", "protein_coding", "TEC". The remaining designations were considered non-coding and include "disrupted domain", "IGC pseudogene", "IGJ pseudogene", "IG pseudogene", "IGV pseudogene", "processed_pseudogene", "transcribed_processed_pseudogene", "transcribed unitary pseudogene", "transcribed_unprocessed_pseudogene", "translated processed pseudogene", "TRJ pseudogene", "unprocessed_pseudogene", "unitary_pseudogene", "3prime overlapping ncrna", "ambiguous orf", "antisense", "antisense RNA", "lincRNA", "ncrna host", "processed_transcript", "sense intronic", "sense overlapping", "lncRNA", "retained_intron", "miRNA", "miRNA_pseudogene", "miscRNA", "miscRNA_pseudogene", "Mt rRNA", "Mt tRNA", "rRNA", "scRNA", "snlRNA", "snoRNA", "snRNA", "tRNA", "tRNA_pseudogene", and "rRNA_pseudogene". A gene was considered non-coding only if all transcripts were non-coding.

## Linkage Disequilibrium (LD)

$R^2$ was calculated for 1000 genomes 'EUR' super population using the 'ld' function from the package 'snpStats' implemented in R and using LDlink.

## Association testing

Generalized linear models to account for LIPA expression (linear) and CAD (additive logistic) using different combinations of variants were compared using ANOVA with a likelihood ratio test (LRT) implemented in R. Gender and age were included as covariates in models explaining LIPA gene expression, while sex, age, hypercholesterolemia, smoking, and number of diseased vessels were included as covariates in models explaining CAD. Bonferroni multiple hypothesis corrected p-values from the LRT comparing models as well as AICs reflecting 'goodness of fit' for individual models are reported.

Differences in LIPA expression between those with or without a history of CAD were calculated using the wilcox.test function in R.

## Allelic Expression Imbalance (AEI)

Allelic RNA expression imbalance (AEI) was assessed using data from GTEx (phe000039.v1. GTEx_v8_ASE.expression-matrixfmt-ase.c1). Candidate variants were subsetted from each individual file, and the deviation of the "REF_RATIO" from the "NULL_RATIO" was plotted for each variant in a given tissue type. Tissue types with 5 or more samples were considered.

## Tissue specific eQTLs

eQTLs reported by GTEx for LIPA were clustered by their LD (R squared calculated from 1000 genomes) using heatmap.2 from the gplots package in R. The p-value reported by GTEx for each eQTL was used to shade the coloring of a tissue specific bar alongside the heatmap using the ColSideColors argument within the heatmap.2 function. In this way tissue specific LD blocks could be visually assessed.

Power calculations based on tissue specific gene expression (median transcripts per million) and sample size were performed for a mock genetic variant assumed to have a MAF of 0.05

and effect size of 40% (i.e. no minor alleles is 20% less than the median tissue specific gene expression and two minor alleles is 20% greater than the median expression). It was assumed 5 million genetic variants were tested. Calculations were executed using the powerEQTL. ANOVA function from the power.EQTL package in R. Results were plotted using barplot function in R.

## Results and discussion

### Expanding candidate gene lists using QTL and position

As many functional variants marked by GWAS likely have regulatory functions affecting RNA expression or processing, the same SNPs appearing in GWAS may also mark expression Quantitative Trait Loci (eQTLs) or splicing Quantitative Trait Loci (sQTLs) for their target gene. To assign GWAS variants to target genes, we determine for each of the GWAS SNPs whether it appears as an eQTL or sQTL reported by GTEx, searching all available tissues. Recognizing that often multiple SNPs exist over a genomic region as significant GWAS variants, we consider each one individually in assigning candidate genes and separately assess concordance. We opt not to use COLOC and other existing tools that search for overlapping signal between GWAS variants and QTLs because they make assumptions about the genetic model that are not in line with the multi-factorial system we test here [25]; namely, these methods assume a single causative variant or that each variant acts independently. Instead, although we recognize it limits the overlap we are able to detect and biases our sample to variants that are ideal markers (i.e. frequent), we search for exact matches between GWAS and QTL marker variants. In addition to evaluating associations with gene expression and splicing, we consider the physical position of each GWAS variant as SNPs within the RNA sequence are expected to impact RNA folding, stability, function, etc. Specifically, we consider the corresponding gene for any transcript that physically overlaps the GWAS variant regardless of strand, thus incorporating coding, non-coding, and antisense genes. Using these three approaches (cis-eQTLs, cis-sQTLs, position), we expand the list of potential candidate genes for the 58 GWAS loci from 75 to 245 (Fig 1A, S1 File, comprehensive table is included in S3 File, S1 Fig).

This phenomenon of expanding a GWAS based candidate gene list by incorporating genes for which the GWAS variant serves as an eQTL/sQTL or on the basis of physical proximity is not unique to the CAD phenotype nor the particular GWAS published by Nikpay et al and their means of annotating genes. We considered two additional phenotypes of insulin resistance and blood pressure with recent large-scale GWAS studies and found these approaches also significantly expanded the range of candidate genes (Fig 1 and S4 File) [26,27].

In an effort to identify those loci where a target gene(s) is clearly supported by functional markers, we consider the agreement between the gene assignment given by GWAS studies and that derived by eQTL and sQTL analysis as well as by physical position. We group each of the 58 GWAS loci as follows: GWAS annotation is supported by QTL-based re-prioritization or position and no other candidate genes are introduced (Tier 1); QTL-based reprioritization or position introduces new genes, while supporting all (Tier2A), some (Tier2B), or none (Tier2C) of the genes annotated by GWAS so that multiple genes are implicated; no eQTLs or sQTLs are identified and no gene or annotated RNA transcript physically overlaps the SNP, accordingly annotation by GWAS is not supported (but also not negated) and no other genes are implicated (Tier 3), see Fig 1, S2 Fig, S3 File.

**Tier 1: No new candidate genes introduced–GWAS annotation supported.** For 7 loci, QTL-based reprioritization and/or position supports the GWAS annotation without introducing new candidate genes, supporting all or some of the gene(s) annotated by GWAS (Table 1).

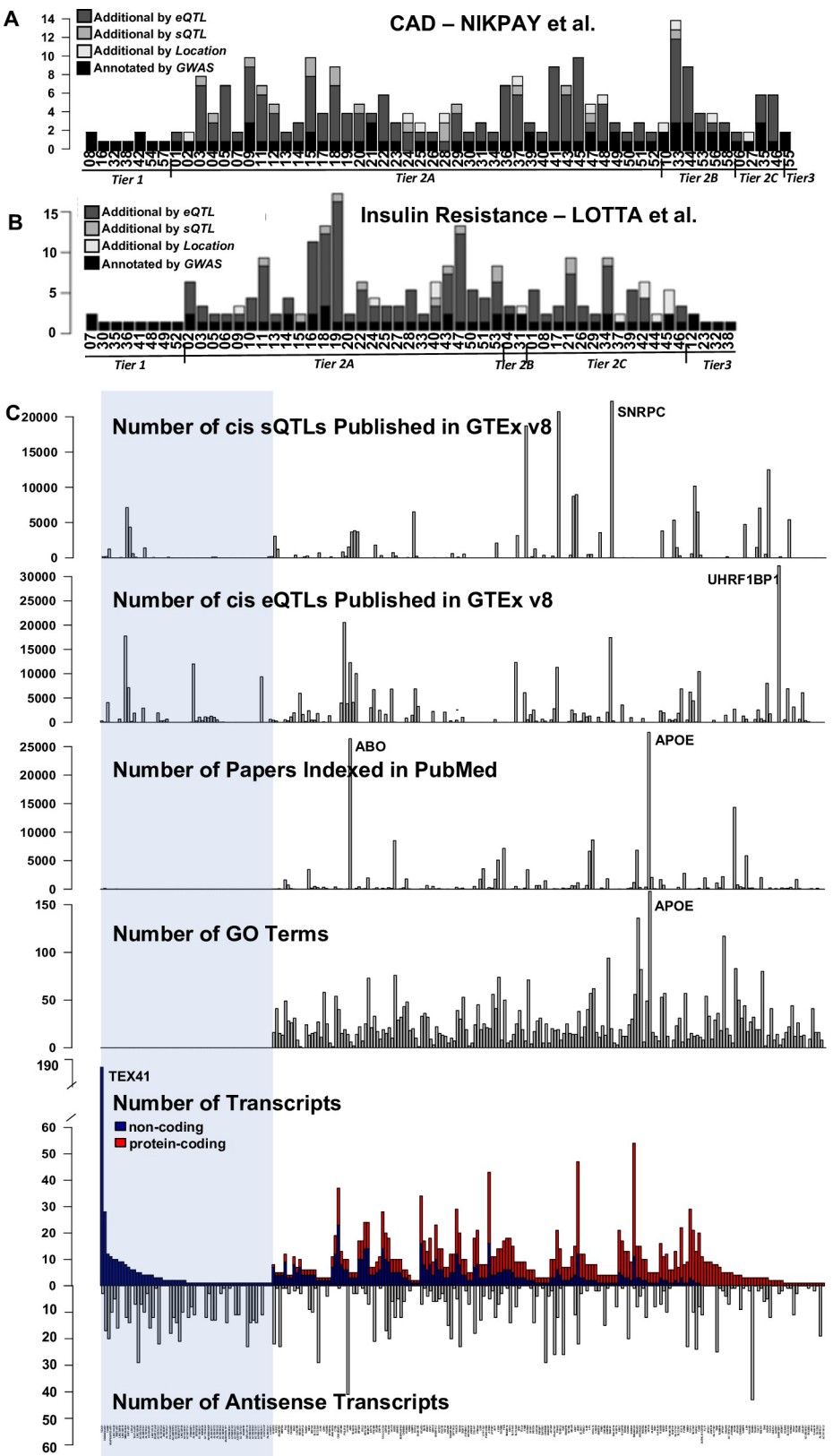

**Fig 1. Summary of CAD GWAS loci.** (**A**) For each of the 58 loci identified by GWAS, number of candidate genes annotated by GWAS and additional genes added by eQTL, then sQTL, and finally position based reprioritization, if

implicating genes other than those annotated previously by GWAS (See S1 Fig for further details about the approach and S3 File for a comprehensive table). Tier 1 (n = 7) denotes those loci where a GWAS annotated gene is supported by QTL-based re-prioritization or position and no other candidate genes are introduced; Tier 2 (n = 50) where QTL-based reprioritization or position introduces new associated genes while supporting all candidates at this locus (Tier2A), only some including the GWAS gene (Tier2B) or new genes except the GWAS genes (Tier2C); and Tier 3 (n = 1) where no eQTLs or sQTLs are identified and no gene physically overlaps the SNP, accordingly annotation by GWAS is not supported and no other genes are implicated. (**B**) Corresponding figure for recent large scale GWAS for insulin resistance. (**C**) For each of the 245 candidate genes displayed along the x-axis (names available in S1 File), the number of transcripts assigned to the gene, the number of antisense transcripts (note: antisense genes are not included among the 245 candidate genes unless their expression is associated with or they physically overlap a GWAS variant), GO terms, Papers indexed in PubMed, *cis*-eQTLs and sQTLs published in v8 of GTEx. Blue bar highlights those genes with only non-coding transcripts.

For four of the loci (16-ADTRP, 32-LIPA, 38-FLT1, 42-FURIN, 8-ABCG8), GWAS annotation of candidate gene assignment is supported by both QTL and position. In one instance, locus 42—rs17514846 (FURIN, FES), more than one gene is annotated by GWAS and supported by our reprioritization. rs17514846, which falls in an intron of FURIN, serves as an eQTL and an sQTL for FES in 23 tissues and an eQTL for FURIN in 3 tissues, two of which (aorta and tibial artery) overlap with FES. In aorta and tibial artery, rs17514846 is associated with decreased expression of FES as opposed to increased expression of FURIN–a possible example of competing interactions between regulatory and promoter regions. Evidence for multiple candidate genes in a locus may represent a paradigm in which a single SNP exerts an impact through more than one gene.

In some instances the same variant in the same tissue is associated with both expression and splicing. For example, rs141244 in blood is associated with increased expression of LIPA and decreased splicing, a scenario that is consistent with greater stability of the un-spliced transcript. Thus, in considering potential mechanisms of action for the variant, it is important to evaluate not only the implications of increased levels of LIPA mRNA, but also increased levels of the un-spliced transcript.

**Tier 2: New candidate genes implicated.** Variants in 50 loci are associated with expression of one or more genes or physically overlap with another gene in addition to all (39 loci), some (7 loci), or none (4 loci) of the genes annotated by GWAS. Loci where additional candidate genes are introduced are classified as Tier 2 (S3 File). Candidate genes for these 50 GWAS loci are expanded by an average of 4.3 genes per locus for a total of 170 genes: 116 from eQTL based reprioritization, 17 from sQTL based reprioritization, 5 from physical position, and 32 from some combination of these features (S3 Fig).

While about a third of the loci (21) have two or fewer candidate genes, others have substantially more: e.g., locus 33—rs12413409 and rs11191416 (CYP17A1-CNNM2-NT5C2) are associated with expression of twelve genes. Importantly, these multi-gene eQTLs cannot be explained solely by co-expression between genes. These eQTLs are often associated with expression of different genes in different tissues. Notably, ncRNAs are candidate genes for 33 of the 58 loci expanded from 6 loci prior to re-prioritization. For no loci are all candidate genes non-coding.

For Tier 2C loci, there is no evidence to support the GWAS annotation. For example, locus 46—rs1122608 and rs56289821 LDLR is annotated by GWAS, a gene well-recognized for its role in lipid metabolism; yet, rs1122608 falls within an intron of SMARCA4 and is both an eQTL and sQTL for SMARCA4 as well as an eQTL for CARM1 and YIPF2 but not LDLR. The alternative SNP identified by GWAS, rs56289821, also does not point to LDLR but rather implicates RGL3, SLC44A2, and again SMARCA4. These 4 Tier 2C loci critically require future work, both mechanistic and computational, to explore relevant gene targets.

**Table 1. Tier 1 CAD GWAS loci.**

| Locus | SNP | OR | Risk Allele (Freq) | Gene | eQTL Tissue(s) | sQTL Tissue(s) | Position |
|---|---|---|---|---|---|---|---|
| 16 | rs6903956 | 1.65[a] (1.44–1.90) | A (0.08[a]) | ADTRP | | Testis | ADTRP (intron) |
| 32 | rs11203042 | 1.04 (1.02–1.06) | T (0.45) | LIPA | Adipose (subq) Adipose (visceral) Colon (transverse) Heart (atrium) Lung Pancreas Skin (sun exp) Spleen Thyroid Blood | Adipose (subq) Fibroblasts Lung | LIPA (intron) |
| 32 | rs1412444 | 1.07 (1.05–1.09) | T (0.37) | LIPA | Adipose (subq) Adipose (visceral) Adrenal Gland Artery (aorta) Brain (cerebellum) Colon (sigmoid) Colon (transverse) Heart (atrium) Heart (LV) Lung Skeletal Muscle Nerve Pancreas Skin (not sun exp) Skin (sun exposed) Spleen Stomach Thyroid Blood | Adipose (subq) Adipose (visceral) Adrenal Gland Artery (aorta) Artery (tibial) Brain (spinal cord) Breast Fibroblasts Lymphocytes Lung Tibial Nerve Pancreas Skin (sun exposed) Small Intestine Spleen Stomach Blood | LIPA (intron) |
| 38 | rs9319428 | 1.04 (1.02–1.06) | A (0.31) | FLT1 | Nerve (tibial) | | FLT1 (intron) |
| 42 | rs17514846 | 1.05 (1.03–1.07) | A (0.44) | FES | Adipose (subq) Adipose (visceral) Adrenal Gland Artery (aorta) Artery (tibial) Fibroblast Colon (transverse) Esophagus (musc.) Heart (atrium) Lung Nerve (tibial) Pancreas Pituitary Prostate Skin (not sun exp) Skin (sun exposed) Stomach Thyroid Blood | Adipose (subq) Adipose (visceral) Artery (aorta) Artery (tibial) Breast Fibroblasts Colon (sigmoid) Esophagus (GEJ) Esophagus (musc.) Heart (atrium) Heart (LV) Lung Salivary Gland Nerve (tibial) Prostate Skin (not sun exp) Skin (sun exposed) Small Intestine Spleen Thyroid Blood | FURIN (intron) |
| | | | | FURIN | Artery (aorta) Artery (tibial) Esophagus | | |
| 54 | rs7212798 | | | BCAS3 | | | BCAS3 (intron) |
| 57 | rs11830157 | | | KSR2 | | | KSR2 (intron) |

(*Continued*)

**Table 1.** (Continued)

| Locus | SNP | OR | Risk Allele (Freq) | Gene | eQTL Tissue(s) | sQTL Tissue(s) | Position |
|---|---|---|---|---|---|---|---|
| 08[b] | rs6544713 | 1.05 (1.03–1.07) | T (0.32) | ABCG8 | Colon (transverse) | | ABCG8 (intron) |

Tissue names in grey font indicate GWAS SNP is associated with a decrease in gene expression (eQTL) or normalized intron-excision ratio (sQTL), while those in black font are associated with increased expression/normalized intron-excision ratio as reported by GTEx.

[a] values reported from original publication [28] in Han Chinese population. rs6903956 was not significant in Nikpay et al. [1].

[b] ABCG8 and ABCG5 were both annotated by GWAS. ABCG5 was not supported by QTL or position.

**Tier 3: No genes implicated.** The remaining GWAS locus, locus 55—rs663129 (MC4R, PMAIP1), classified as Tier 3, did not show any association with expression of nearby genes and is not physically overlapping any transcripts (S3 File). This locus and 3 others (locus 27—rs2954029 (TRIB1), locus 54—rs7212798 (BCAS3), and locus 57—rs11830157 (KSR2)) that are without any eQTL associations may have more subtle or context-dependent effects on gene expression that remain undetectable in GTEx. In particular, non-polyadenylated transcripts are not in GTEx as poly-dT priming was used, leaving countless ncRNAs as additional candidates. Furthermore, these SNPs may affect gene expression in *trans* (although we do not find such evidence in the GTEx *trans*-eQTL dataset) or exert their effect without altering RNA levels measured by RNAseq (*e.g.* by controlling the chromatin structure or co-translationally alter RNA modifications). Additionally, variants affecting RNA functions and processing (structural RNA SNPs) [8,29], may not be visible as eQTLs, or they may selectively affect translation by changing polysomal loading [30].

Given GWAS variants are expected to mark functional variants rather than themselves being functional, we test SNPs within a 1MB window in LD ($R^2 > 0.8$) with each of the 4 GWAS variants lacking annotations, expanding the number of SNPs to 200. Using this approach, we find significant eQTLs, but no significant sQTLs, for three of the four loci. For locus 57, we were unable to find additional candidate SNPs with an $R^2 > 0.8$ to mark the haplotype.

## Survey of CAD GWAS candidate genes

The genomic loci for each these 245 candidate genes often harbor multiple protein-coding and non-coding transcripts arranged on both the sense and antisense strands (S2 File). They express an average of 9 transcripts per gene and a maximum of 189 (TEX41- locus 10—rs2252641, rs17678683), with 47% of all transcripts being non-coding (Fig 1B). More than half of the gene loci (161) also contain one or more antisense genes (i.e., located on the opposite strand and overlapping).

With a median of 26 publications and a maximum of 27,497 (APOE), only a handful of these 245 candidate genes have been well studied to date (Fig 1B). Twenty percent (51) of genes do not have a single paper indexed in PubMed. There are on average 20 gene ontology (GO) terms, which are manually curated based on the literature, assigned to each gene; however, 62 (25%) of the candidate genes have no associated GO terms. We find those genes without GO terms and with limited publications do not have fewer markers of functionality (eQTLs, splicing QTLs, etc.), but are almost exclusively non-coding, indicative of a recognized bias in the literature toward protein-coding genes (Fig 1).

Each implicated locus displays an astoundingly complex architecture with multiple candidate genes implicated by RNA expression and physical location, each with a number of overlapping coding and non-coding transcripts including those in antisense orientation. The complexity of these loci emphasizes the need for targeted molecular studies and computational

approaches to determine the relevant gene and transcript(s). The distribution of PubMed articles and GO ids across candidate genes suggests that this targeted work has touched on only a handful of genes thus far, with more recent studies beginning to focus on 'neglected' CAD candidate genes [31].

## Allelic RNA expression imbalance reveals functional variation

To evaluate potential functionality for each of the 58 GWAS loci, we ask whether each candidate SNP is associated with allelic expression imbalance (AEI), a specific indicator of *cis*-acting regulatory variation. By comparing expression of the two alleles at a heterozygous variant, various external/*trans*-acting influences on gene expression are shared and the *cis*-acting effect of the heterozygous variant can be isolated. In the absence of a functional variant altering RNA expression, the anticipated distribution between the alleles is 0.5 (ratio = 1) [8,32].

Using data released by GTEx, we evaluate AEI at each of 104 candidate variants across 54 tissue types. Only 55 of the SNPs are represented in the data. The remainder likely are in intergenic regions and poorly captured by RNA sequencing, while obtaining accurate AEI ratios requires rather robust expression (>30 RPM) [33]. Of the 55, many are present in only a few samples making it difficult to infer differential expression. However, several SNPs show surprisingly robust data–thousands of samples and dozens-hundreds of counts for each allele. A majority of these SNPs fail to reveal allelic expression imbalance, with near normal distribution of deviation from the expected ratio, suggesting no correlation between the GWAS variant and allelic expression imbalance. This implies that the GWAS candidate SNPs represented in the data are actually relatively poor markers for functional *cis*-acting variants in the locus; however, splicing events generating RNA isoforms with similar turnover are one example where allelic expression imbalance would fail.

A number of SNPs do display consistent allelic expression imbalance (Fig 2). Locus 3 – rs7528419 (SORT1), which falls in the 3'UTR of CELSR2 exhibits AEI in 53/57 liver samples. Overall low expression of CELSR2 in liver tissue, however means that these ratios are for the most part based on low coverage (median total count 13). Despite the relative consistency from sample to sample, large allelic ratios derived from relatively low counts, as observed here, raises suspicion for systemic sources of bias, *e.g.* preferential amplification of one allele. To evaluate this further, we considered allelic ratios at nearby SNPs in strong LD ($R^2 > 0.9$) and weak LD ($R^2 < 0.1$). As these SNPs are co-located, systemic sources of bias should affect all SNPs in the locus while 'true' biological AEI would be expected only for those variants in strong LD with a functional SNP. We observe AEI for those SNPs in strong LD with the GWAS marker, but not for those in the same region in weak LD, a pattern that is suggestive of 'true' biological AEI and a functional cis-acting variant.

Importantly, even one sample without AEI suggests the variant might itself not be functional but rather in high LD with a functional variant and serving as a marker. With only a few samples not exhibiting AEI, rs7528419 can be considered an excellent marker in tight LD with the functional SNP. Furthermore, that this pattern is only found in liver suggests that the regulatory variant is tissue specific. In contrast, the bidirectional ratios observed in adipose tissue suggests that the SNP is not in tight LD with a variant that is functional in adipose tissue.

The Locus 14 SNP rs72689147 (*GUCY1A3*), which falls within an intron of GUCY1A3, exhibits AEI in 114/121 samples across 10 different tissues. Again, this SNP does not appear to be functional as not all samples display AEI, but it is a robust marker. While located in an intron, expression is sufficient to extract allelic ratios; as these are consistently below unity, this results suggests a gain of function.

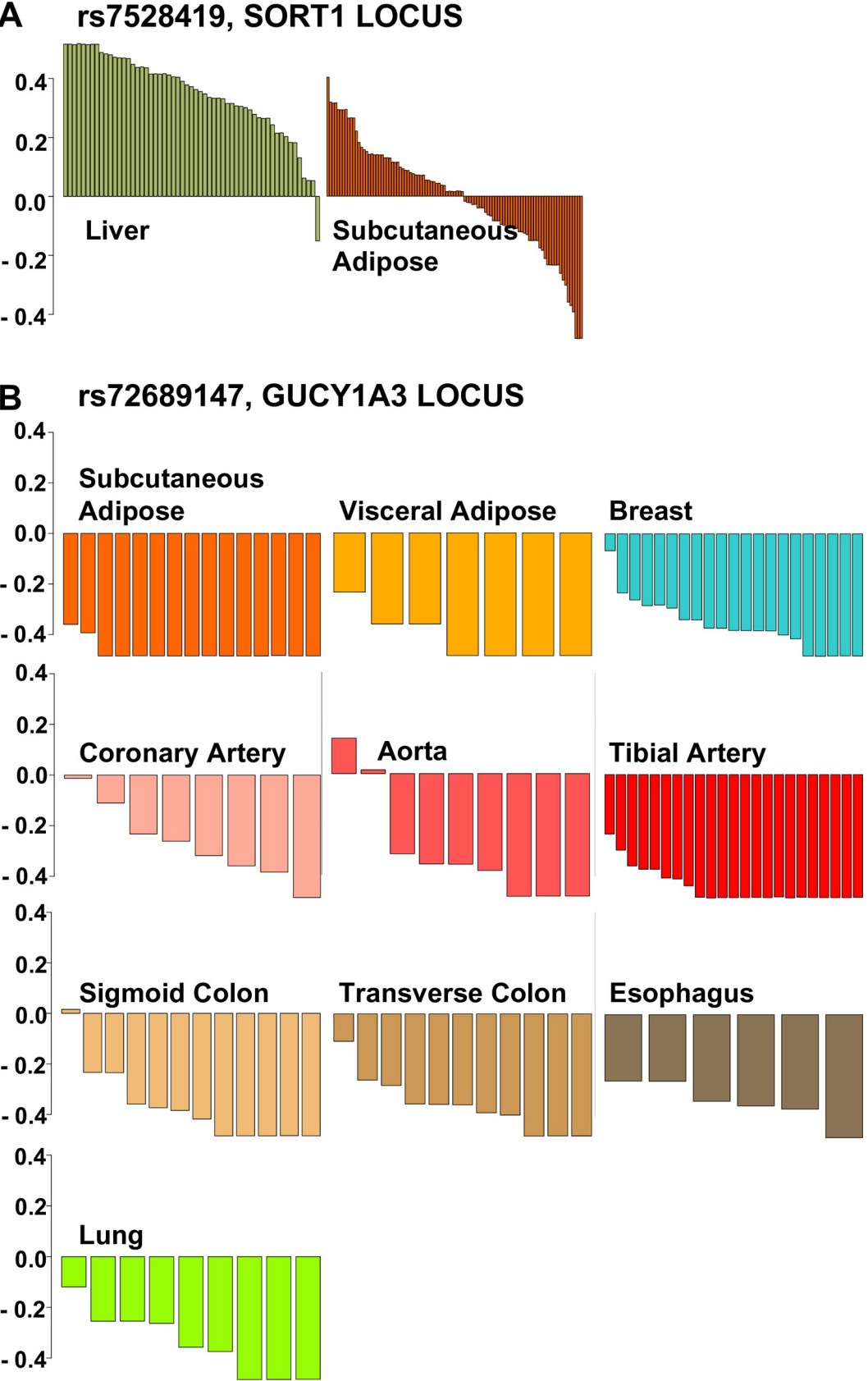

**Fig 2. Allelic expression imbalance at GWAS variants mark functional SNPs.** Deviation in the observed from the expected ratio for individuals heterozygous for given GWAS variant. (A) Locus 3 –rs7528419 (SORT1) exhibits AEI in 53/57 liver samples. Subcutaneous adipose, also shown, demonstrates near normal distribution of deviation from the expected allelic ratio and is representative of the 46 other tissues with at least 5 samples. (B) Locus 14—rs72689147 (GUCY1A3) exhibits AEI in 114/121 samples across 10 different tissues.

## Resolving number of signals in a locus using LD

Focusing on the 7 loci where eQTL-based reprioritization pointed to a single gene as well as the two examples of AEI discussed above, we find dozens of other significant eQTLs for each gene. To determine whether these eQTLs represent one or more functional variants, we plot the effect size (beta) of the variant on RNA expression for each eQTL against its LD ($R^2$) with the top scoring (most significant p-value) eQTL in each tissue where eQTLs are detectable. Assuming one functional variant in the locus, the beta for each eQTL should correlate with its $R^2$ relative to the highest scoring SNP [34].

This approach reveals that the observed eQTLs for a gene often represent more than one regulatory variant, with the exception of FLT1 in Tibial Nerve–represented by only one cluster of variants marked by the GWAS SNP (Fig 3). This result is critical to the correct interpretation of GWAS that would otherwise rely on a single variant rather than considering the combined effect of more than one causative variant.

As an example, we consider the number of distinct eQTLs needed to maximally account for LIPA expression in blood. The most significant eQTL consists of a group of SNPs in high LD marked by the GWAS variant (red dot in Fig 3), while two additional clusters of SNPs (marked by rs1051338 and rs2250781) have equally or even more robust beta and p-values but show relatively poor linkage with the GWAS cluster ($R^2 \sim 0.5$) (Fig 3B). These SNPs are more significant eQTLs than predicted by their LD with the trait-associated variant and may mark additional functional variants in the locus. To test the significance of any additional regulatory variants, we used a separate dataset (CATHGEN) to evaluate whether including an additional marker variant in a regression model improves the ability to account for LIPA expression in blood. Including additional markers improved the eQTL model, while adding a marker in strong LD with the original variant did not (Table 2), indicating there are likely multiple functional variants, incompletely represented by the GWAS variant alone, that contribute to LIPA expression in blood.

Testing these additional variants with CAD instead of LIPA expression did not yield significant associations (Table 2). However, LIPA expression itself is not associated with CAD except when rs1412444 is homozygous minor, which may explain the discrepancy. In looking separately at the associations between the GWAS variant and LIPA expression and the GWAS variant and CAD, we find that rs1412444 is associated with increased risk of CAD and increased expression of LIPA, but counterintuitively those with two minor alleles and CAD exhibit lower rather than higher expression, a pattern that also holds in GTEx although it is only statistically significant in CATHGEN (Figs 4 and S4).

Absence of LIPA results in Wolman disease, characterized by lipid deposits and early onset CAD due to inability to break down lipids in lysosomes and subsequent upregulation of cholesterol production by the liver [35]. Here, congruent with this rare genetic disease, we find decreased *LIPA* expression associated with CAD. Unexpectedly this is observed when the GWAS based variant (rs142444), associated with increased *LIPA* expression, is homozygous minor, implying existence of an additional factor associated with the GWAS variant that interrupts the linear relationship between the number of rs142444 minor alleles and LIPA expression.

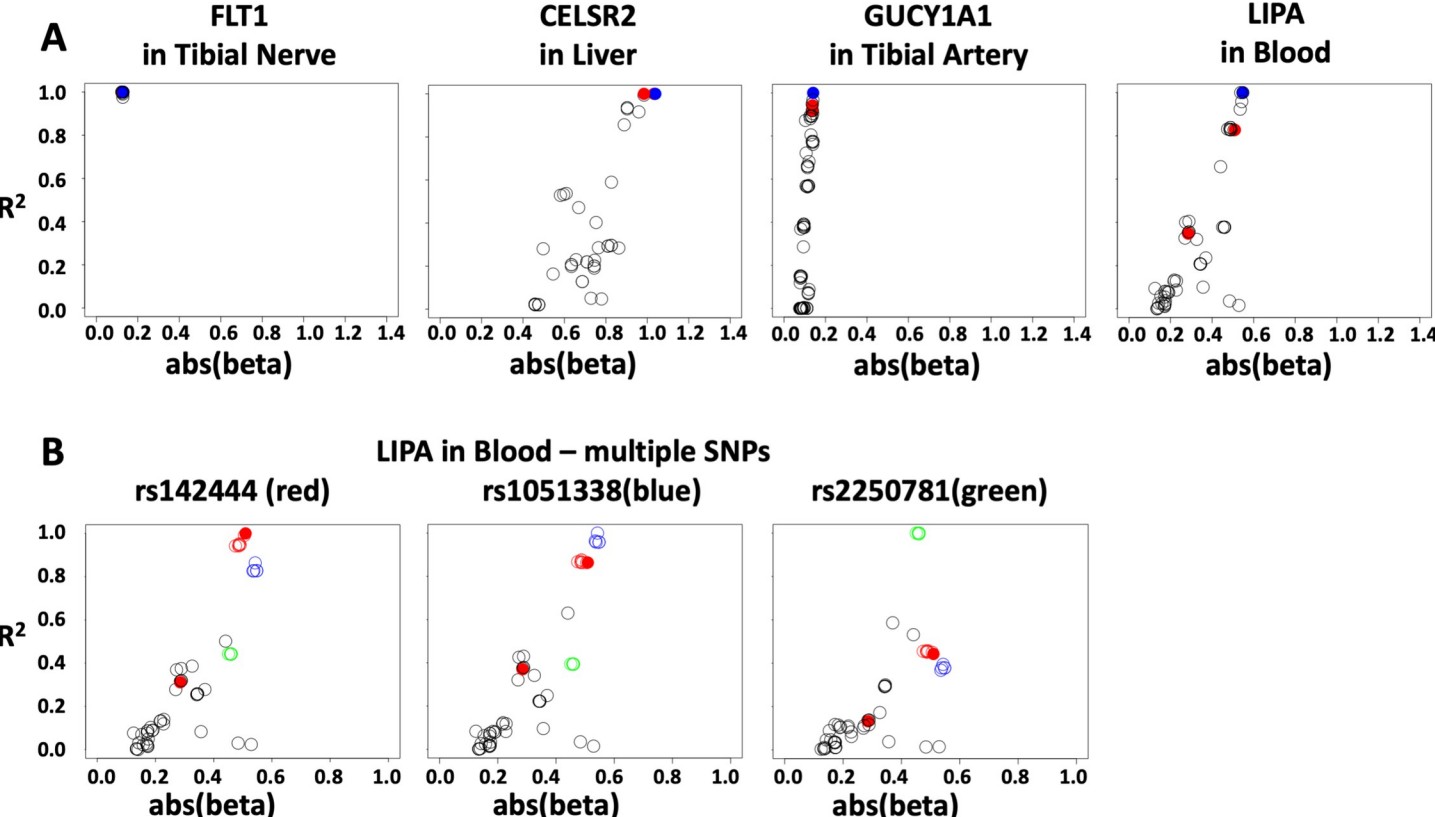

**Fig 3. Number of eQTL signals.** Correlation plots show absolute value of beta for variant effects on RNA expression versus $R^2$ with the top eQTL (most significant p-value), including all significant eQTLs in the given gene-tissue combination. Blue dots represent the top eQTL (most significant p-value), red dots represent GWAS variant(s). (A) FLT1 in Tibial Nerve: eQTLs are accounted for by a single eQTL marked by the GWAS variant (all eQTLs display a linear correlation with $R^2$). CELSR2 (liver), GUCY1A3 (tibial artery), and LIPA (blood), correlation between beta and $R^2$ suggests multiple functional variants. (B) At least three distinct LD blocks represented by LIPA eQTLs in whole blood. Correlations are shown left to right between the absolute value of beta and $R^2$ with rs142444 (GWAS SNP), rs1051338, or rs2250781. Tightly linked SNPs (D' > 0.9; $R^2 > 0.9$) are shown in the same color.

## Context–tissue specific eQTLs

Genetic variation exists and functions within a context–the surrounding sequence, the tissue type and its preferred transcription factors, etc. In an effort to resolve the functional variation behind statistical associations observed in GWAS, it is essential to consider these contexts. As highlighted by the tissue specific AEI patterns above, if these relationships are not considered in a context specific manner (*e.g.*, on a tissue by tissue basis), many robust effects will remain hidden. In an effort to evaluate some of these contextual features, we consider tissue specific eQTLs.

eQTL analysis may focus the search on a relevant tissue. However, eQTLs are detectable only where expression and sample size are sufficiently high; accordingly tissue-specific differences in eQTLs reflect overall patterns of tissue selective expression and sample size, in addition to the influence of genetic variation (S5 Fig).

To consider how eQTLs for a given gene compare across different tissues, we cluster genome-wide significant eQTLs reported by GTEx for LIPA in a heatmap organized by their pairwise LD ($R^2$), using a colored bar at the top of the heatmap to denote tissue type (Fig 5A). eQTL SNPs generally cluster by tissue, suggesting distinct regulatory variants in different tissues. However, there are two LD blocks that contain eQTLs in more than half of tissues

**Table 2. Assessing multiple regulatory variants for LIPA.**

| Variable of interest | ANOVA p-value | Model 1 | AIC | Model 2 | AIC |
|---|---|---|---|---|---|
| rs1412444 | 8.8e-16 | XP ~ sex + age | 3310 | XP ~ rs1412444 + sex + age | 1778 |
| rs13332328 | 8.8e-16 | XP ~ sex + age | 3310 | XP ~ rs13332328 + sex + age | 1782 |
| rs1051338 | 8.8e-16 | XP ~ sex + age | 3310 | XP ~ rs1051338 + sex + age | 1779 |
| rs2250781 | 8.8e-16 | XP ~ sex + age | 3310 | XP ~ rs2250781 + sex + age | 1800 |
| rs1412444 *in context of* rs13332328 | 1.0 | XP ~ rs1412444 + sex + age | 1778 | XP ~ rs1412444 + rs13332328+ sex + age | 1781 |
| rs1412444 *in context of* rs1051338 | 0.23 | XP ~ rs1412444 + sex + age | 1778 | XP ~ rs1412444 + rs1051338 + sex + age | 1777 |
| rs1412444 *in context of* rs2250781 | 0.04 | XP ~ rs1412444 + sex + age | 1778 | XP ~ rs1412444 + rs2250781 + sex + age | 1773 |
| rs1412444 & rs2250781 *in context of* rs1051338 | 0.19 | XP ~ rs1412444 + rs2250781 + sex + age | 1773 | XP ~ rs1412444 + rs2250781 + rs1051338 + sex + age | 1773 |
| rs1412444 & rs1051338 *in context of* rs2250781 | 0.04 | XP ~ rs1412444 + rs1051338 + sex + age | 1777 | XP ~ rs1412444 + rs1051338 + rs2250781 + sex + age | 1773 |
| rs1412444 | 1e-3 | CAD ~ covariates | 387.7 | CAD ~ rs1412444 + covariates | 385.1 |
| rs13332328 | 1e-3 | CAD ~ covariates | 387.7 | CAD ~ rs13332328 +covariates | 385.3 |
| rs1051338 | 6e-4 | CAD ~ covariates | 387.7 | CAD ~ rs1051338 + covariates | 384.3 |
| rs2250781 | 4e-4 | CAD ~ covariates | 387.7 | CAD ~ rs2250781 + covariates | 383.5 |
| rs1412444 *in context of* rs13332328 | 0.79 | CAD ~ rs1412444 + covariates | 385.1 | CAD ~ rs1412444 + rs13332328+ covariates | 386 |
| rs1412444 *in context of* rs1051338 | 0.36 | CAD ~ rs1412444 + covariates | 385.1 | CAD ~ rs1412444 + rs1051338 + covariates | 384.5 |
| rs1412444 *in context of* rs2250781 | 0.56 | CAD ~ rs1412444 + covariates | 385.1 | CAD ~ rs1412444 + rs2250781 + covariates | 385.9 |
| rs1412444 & rs2250781 *in context of* rs1051338 | 0.11 | CAD ~ rs1412444 + rs2250781 + covariates | 385.9 | CAD ~ rs1412444 + rs2250781 + rs1051338 + covariates | 385.1 |
| rs1412444 & rs1051338 *in context of* rs2250781 | 0.17 | CAD ~ rs1412444 + rs1051338 + covariates | 384.5 | CAD ~ rs1412444 + rs1051338 + rs2250781 + covariates | 385.1 |

ANOVA comparing generalized linear models with different SNP combinations accounting for LIPA expression and CAD (defined as history of myocardial infarction and/or >50% stenosis of vessel). Covariates in CATHGEN include sex, age, hypercholesterolemia, smoking, and number of diseased vessels.

indicative of genetic variation that acts across different tissue types. Variants detected by GWAS for LIPA appear as a significant eQTLs in a subset of tissues (Table 1), some of which fit with our understanding of CAD pathology (heart, aorta, adipose), others suggest as yet unexplained biological consequences (spleen, pancreas) or pleiotropic effects.

## Conclusions

We consider each of 58 loci implicated in CAD by GWAS to understand the biological meaning of the underlying statistical associations. In evaluating each of these loci, we find numerous candidate genes that were not included in the original annotation by GWAS. Many of these are non-coding. Non-coding RNAs, now well-recognized for their role as regulators, have historically been dismissed and continue to be difficult to study, a trend that is apparent in their poor representation in the literature, among GO annotations, and as annotated by GWAS [36]. We find no evidence to suggest these non-coding RNAs are less likely to account for the observed associations in GWAS and would advocate for their inclusion in further mechanistic and computational work examining these loci. In addition to broadening candidate gene lists to include non-coding transcripts, we would urge reconsideration of current assignments, especially for those loci categorized as Tier2C where expression, splicing, and physical position do not support the gene annotated by GWAS. LDLR is a particularly prominent example. Given our understanding of the critical role lipid metabolism plays in CAD, it is counterintuitive not to assign a CAD GWAS variant to LDLR when it lies within 15kb of the *LDLR* locus [37]. However, RNA expression and splicing data do not support this annotation, instead

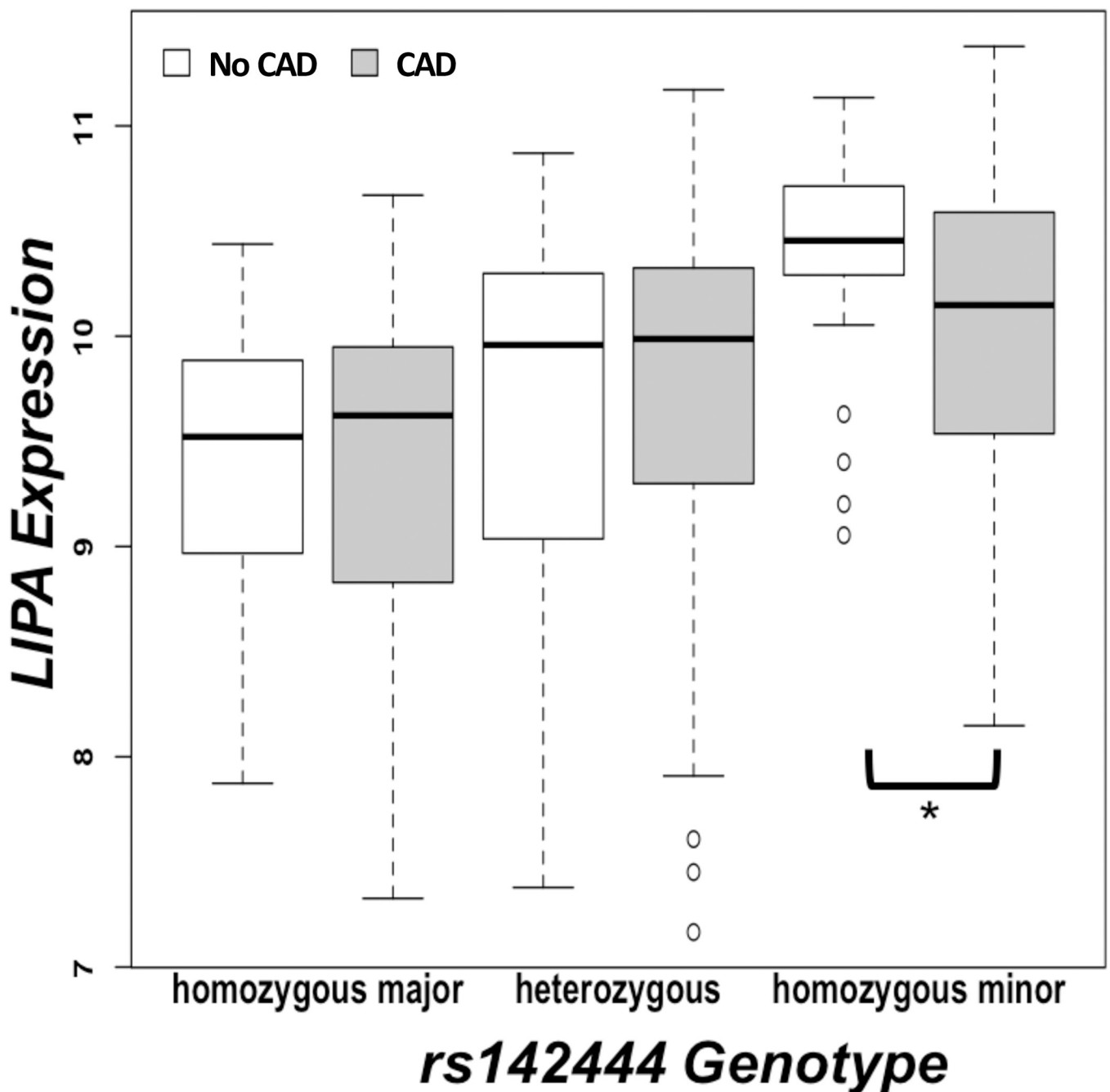

**Fig 4. LIPA expression, CAD, and genotype.** Comparison of LIPA expression in CATHGEN for those with and without CAD based on rs142444 genotype. LIPA exhibits higher expression only in those without CAD in the homozygous minor group (p-value = 0.02).

supporting the notion that such genetic variation affects the function of other nearby genes including SMARCA4, CARM1, YIPF2, RGL3, SLC44A2 [30].

Using allelic ratios built from tissue-specific RNA sequencing data available through GTEx, we were able to identify two loci where the GWAS variant served as a robust marker for a functional *cis*-acting regulatory variant. Locus 3 –rs7528419 (SORT1) falls in the 3'UTR of CELSR2, exhibits AEI exclusively in liver, and is in nearly perfect LD with rs12740374 which

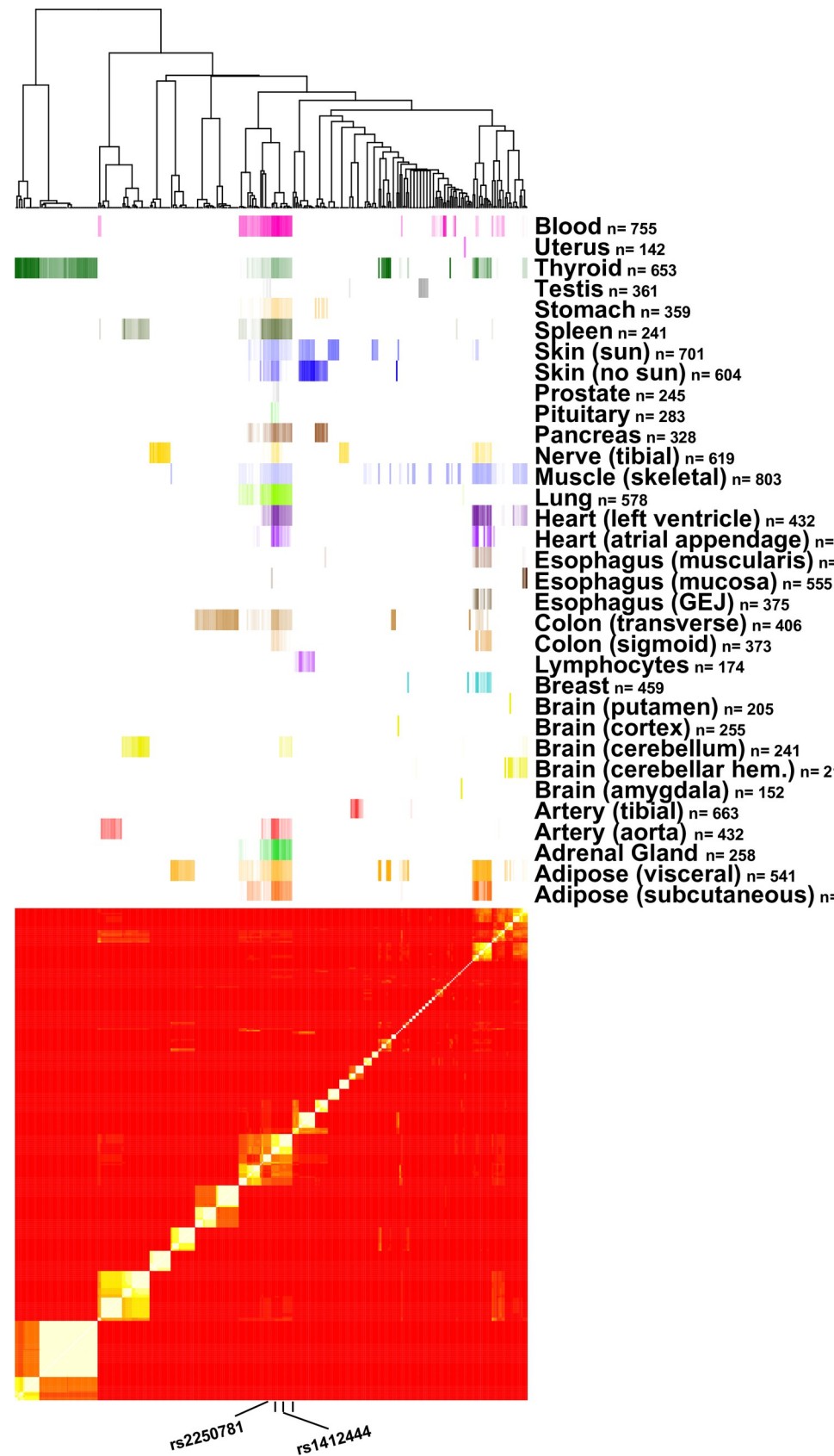

**Fig 5. Tissue specific eQTLs for LIPA.** Heatmap of LD for those SNPs reported by GTEx as genome-wide significant eQTLs for LIPA. Lighter-colored squares in the heatmap represent LD blocks, with SNPs clustered by $R^2$ and not by genomic position. Colored bars at top eQTLs in each tissue with more significant p-values denoted by darker color.

was shown by Musunuru et al. through a series of molecular experiments to create a C/EBP binding site increasing expression of SORT1, a multiligand sorting receptor which they concomitantly showed to be associated with LDL-C and VLDL levels [38]. This work revealed a single functional variant for a single target gene with a substantial effect size, the authors estimated a ~40% difference in MI risk. Our work suggests additional eQTLs not explained by their LD with the LD block marked by GWAS variant rs7528419. As we begin to identify functional variation behind GWAS associations, an important next step will be resolving additional functional variants within the loci that may modify these associations and better account for disease risk [39].

This work emphasizes that the linear presentation of GWAS results as a single variant tied to a single gene fails to capture the complexity of these loci. Many loci contain several SNPs identified by GWAS, and for each of these, multiple candidate genes are implicated by RNA expression and splicing associations as well as physical proximity. LD alone rarely accounts for the observed eQTLs, suggesting multiple functional variants within these loci. Although some GWAS associations may ultimately implicate single variants that alter expression of individual genes, this work indicates that true genetic effect size of a gene locus is accounted for by a multi-factorial system that allows for multiple functional variants regulating one or more genes. Important next steps in accounting for the genetic basis of disease will be in establishing causality for genetic variation, which even with computational efforts such as these to direct our understanding will require molecular biology experimentation to definitely address. It will also require looking beyond single nucleotide polymorphisms to copy number variation, methylation, and other forms of genetic variation, which have been shown to have considerable impacts on disease risk. Ultimately considering how functional variation of all kinds can in combination can be used to predict disease risk will likely machine learning approaches that can more effectively incorporate multi-factorial data [40,41]. The approach presented here must be expanded to include functional variants that are undetectable by RNAseq of whole tissues, including cell type specific expression, effect on RNA-protein interactions, distribution in sub-cellular domains, alteration of translational processes, and of course variants that change protein functions.

## Supporting information

**S1 Fig. Expanding candidate genes process.** Flowchart portraying process of expanding candidate gene list from 75 to 245 using eQTL, sQTL, and physical position.
(TIF)

**S2 Fig. Tier assignment process.** Flowchart portraying process of assigning tiers to CAD GWAS loci.
(TIF)

**S3 Fig. eQTL, sQTL, position Venn diagram.** Venn diagram showing overlap in candidate genes derived from eQTL, sQTL, and position-based re-prioritization.
(TIF)

**S4 Fig. LIPA expression, CAD, and genotype in GTEx.** Comparison of LIPA expression in GTEx for those with and without heart disease based on rs142444 genotype. LIPA exhibits higher expression in those without heart disease only in the homozygous minor group (p

value = 0.22).
(TIF)

**S5 Fig. Power calculations for tissue specific eQTLs.** Barplot displays power to detect a hypothetical LIPA eQTL with minor allele frequency 0.05 and effect size 40% (i.e. no minor alleles is 20% less than the median tissue specific gene expression and two minor alleles is 20% greater than the median expression) across different tissue types. About half of the tissues have greater than 80% power to detect such a variant.
(TIF)

**S1 File. Fig 1 Gene names.** Gene names corresponding to bar plot presented in Fig 1B.
(DOCX)

**S2 File. Example locus.** Example of a locus (*LIPA*) implicated by GWAS taken from ensemble. org. There are numerous annotated protein-coding and non-coding transcripts in close proximity and overlapping one another.
(DOCX)

**S3 File. 58 CAD GWAS loci.** Table of 58 GWAS loci including tier designation, SNPs considered, GWAS annotation, and genes introduced by eQTL, sQTL, and position. Also includes additional text describing Tier 3 loci and the expanded search for candidate genes.
(DOCX)

**S4 File. Additional phenotype–blood pressure.** Bar charts showing the distribution of tier assignments for each of the GWA studies considered. Tier assignments for each of the 903 loci identified in a recent blood pressure GWAS [27].
(PDF)

## Acknowledgments

We would like to acknowledge the publicly available datasets that this work is based on, the GTEx Project and CATHGEN. Please see https://www.ncbi.nlm.nih.gov/projects/gap/cgi-bin/study.cgi?study_id=phs000424.v8.p2 and https://www.ncbi.nlm.nih.gov/projects/gap/cgi-bin/study.cgi?study_id=phs000703.v1.p1 for more details about how these studies were made possible.

## Author Contributions

**Conceptualization:** Katherine Hartmann, Wolfgang Sadee.

**Data curation:** Katherine Hartmann.

**Formal analysis:** Katherine Hartmann, Michał Seweryn.

**Funding acquisition:** Wolfgang Sadee.

**Methodology:** Michał Seweryn.

**Writing – original draft:** Katherine Hartmann.

**Writing – review & editing:** Katherine Hartmann, Michał Seweryn, Wolfgang Sadee.

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
