## [Decision Letter · Decision Letter 0]

6 Apr 2021

PONE-D-20-39555

Interpreting coronary artery disease GWAS results: A functional genomics approach assessing biological significance

PLOS ONE

Dear Dr. Hartmann,

Thank you for submitting your manuscript to PLOS ONE. After careful consideration, we feel that it has merit but does not fully meet PLOS ONE’s publication criteria as it currently stands. Therefore, we invite you to submit a revised version of the manuscript that addresses the points raised during the review process.

We look forward to receiving your revised manuscript.

Kind regards,

Mingqing Xu

Academic Editor

PLOS ONE

Journal Requirements:

Thank you for stating the following in the Acknowledgments Section of your manuscript:

This study was supported by National Institutes of Health National Institute of

General Medical Science Pharmacogenetics Research Network [Grant U01 GM092655]

and the National Center for Advancing Translational Sciences [TL1 TR001069].

The GTEx Project was supported by the Common Fund of the Office of the

Director of the National Institutes of Health, and by NCI, NHGRI, NHLBI, NIDA, NIMH,

and NINDS. The data used for the analyses described in this manuscript were obtained

from: the GTEx Portal and dbGaP accession number phs000424.

For CATHGEN, clinical data originated from the 604 Duke Databank for Cardiovascular

Disease (DDCD) and biological samples originated from the Duke Cardiac CATHeterization

 (CATHGEN) study. Funding support for the Genetic Mediators of Metabolic CVD Risk was

provided by NHLBI grant RC2 HL101621 (William E. Kraus). The data used for the analyses

described in this manuscript were obtained from the dbGaP accession number

phs0000703.v1.p1.

Computing time provided by the Ohio Supercomputer Center, GRANT #: PAS0885-2

and the Prometheus Cyfronet AGH.

This study was supported by National Institutes of Health National Institute of General Medical Science Pharmacogenetics Research Network [Grant U01 GM092655 awarded to WS] https://www.nigms.nih.gov/ and the National Center for Advancing Translational Sciences [TL1 TR001069 awarded to KH] https://ncats.nih.gov/.

Please revise according to the reviewer's comment for re-submission.

Reviewers' comments:

Reviewer's Responses to Questions

**Comments to the Author**

1. Is the manuscript technically sound, and do the data support the conclusions?

Reviewer #1: Yes

Reviewer #2: No

2. Has the statistical analysis been performed appropriately and rigorously? 

Reviewer #1: I Don't Know

Reviewer #2: No

3. Have the authors made all data underlying the findings in their manuscript fully available?

Reviewer #1: Yes

Reviewer #2: Yes

4. Is the manuscript presented in an intelligible fashion and written in standard English?

Reviewer #1: Yes

Reviewer #2: Yes

5. Review Comments to the Author

Reviewer #1: The manuscript “Interpreting coronary artery disease GWAS results: A functional genomics approach assessing biological significance” has an overall goal of addressing an important question, namely, when GWAS identifies multiple variants in a given locus is it due to the underlying LD or because of multiple variants with independent/only partially related functional effects? It also seeks to determine if incorporating tissue-wide eQTLs and splicing QTLs (sQTLs) is a “better” way to determine the underlying gene for a given non-coding locus. All of these efforts as a concept are important as we think about the non-coding genome and primarily non-coding variants identified from GWAS.

The strengths of the study are the overall conceptual framework posed (with caveat above that the analyses presented only take us so far); the careful incorporation of GTEx data coupled with individual level data obtained from the CATHGEN study through dbGaP; and the careful thought put into the genomic interpretations.

But while the overall conceptual framework and potential hypotheses are important, the manuscript’s impact is weakened due to the overall descriptive data presented; the expansive results section for primarily relatively straightforward bioinformatics work that would be done when evaluating any genetic variant; and the lack of functional validation of the proposed model outside of eQTLs to show that this approach is “better” than the comparator approach (which in this paper is just comparison to annotation in the Nikolay et al paper). In fact, for many of their SNPs, a simple search in dbSNP/UCSC confirms the gene that the SNP is in (not sure why the Nikolay paper annotated it differently). As such, the comparator being the Nikolay paper doesn’t help us determine if this method is “better”.

More minor issues include:

1) To address the above issue of the comparator being “GWAS annotated” but from a single paper, the authors should consider other phenotypes to perform these analyses;

2) The results section is very long and could be summarized more succinctly;

3) Throughout, there needs to be more formal statistics done (and more statistics methods)…for example, in Table 2, ANOVA does not make sense for the model for association between SNPs and MI; what are p-values/effect sizes/directionality for eQTLs, for “heat map” for clustering and showing that some variants have eQTLs in multiple tissues, etc.

4) For the LIPA analyses, why use MI when the original variants are CAD variants?

5) The LIPA expression models are interesting and statistically a good way to further validate that multiple SNPs in a locus have independent effects and should be done across all the loci.

6) For the LIPA expression models, need more formal statistics to show that adding SNPs improves models for expression (i.e. AIC, BIC, etc.)

7) Overall, the manuscript could use an eye towards editorial improvement (uses colloquial language in several places like calling them “GWAS hits”, there are grammatical errors, results section is densely presented; catheterization in methods is spelled incorrectly; Wilcoxon is spelled “wilcoxin”; the test in R is wilcox.test not “wilcoxin.test”, etc.)

8) How do the authors interpret the paradoxical results for rs1412444 on LIPA expression and MI?

9) How were eQTLs in GTEx defined? (again gets back to inclusion of more formal statistics). It’s hard to determine if some of the differences between variants/across tissues could just be variations around statistical significance without these more granular results.

Reviewer #2: In the manuscript entitled “Interpreting coronary artery disease GWAS results: A functional genomics approach assessing biological significance”, the authors explored the complexity of each of CAD loci by use of tissue specific RNA sequencing data from GTEx to identify genes that exhibit altered expression patterns in the context of GWAS significant loci,and expanded the list of candidate genes from the 75 currently annotated by GWAS to 245.

The following papers can be cited and followed for the meta-analytic procedures(if the data is not enough available, at least DISCUSSION should be added as the LIMITATION of this study with enough citation to support the viewpoints):

Ref 1: Wu Y, et al. Multi-trait analysis for genome-wide association study of five psychiatric disorders. Transl Psychiatry. 2020 Jun 30;10(1):209.

Ref 2:Jiang L, et al. Sex-Specific Association of Circulating Ferritin Level and Risk of Type 2 Diabetes: A Dose-Response Meta-Analysis of Prospective Studies..J Clin Endocrinol Metab. 2019 Oct 1;104(10):4539-4551.

Ref 3: Xu M, et al. Quantitative assessment of the effect of angiotensinogen gene polymorphisms on the risk of coronary heart disease. Circulation. 2007 Sep 18;116(12):1356-66

Trans-ethnitic and trans-trait meta-analysis of cardiometabolic traitscan be referred to Ref 1.

Subgroup analyses based on sex, age, race, gene dosage can be referred to Ref 2. and3

Integrating GWAS signals with eQTL from GTEX or pQTLs is a good strategy to exploring the causality of the genetic varients in the development of cardiometabolic traits. But I strong suggest to do causal inference analysis to see if the GWAS signals are causally triggering the develop,ent of CAD through mediating the expression of given genes in specific tissues.

In addition, the significantly associated SNPs may be used to predict disease susceptibility in the context of its influence of gene expression,therefore, the authors may explore the possibility to conduct a machine-learning model to predict CAD risk or cardiometalobic traits based these significant SNPs. For this reason, the authors may cite the following papers to follow these references’ procedure to construct a standard prediction model based on the significant SNPs (probably include the gene expression information). Deep learning is a hot topic in dissecting the genome variants’ roles in the phenome. Especially deep learning method is a very promising way to predict disease risk based on clinical information and genetic biomarkers(If deep learning can not be used, please discuss as the LIMITATION of this study with enough citation to support the viewpoints).

Ref 4:Yu H, et al. LEPR hypomethylation is significantly associated with gastric cancer in males.Exp Mol Pathol. 2020 Oct;116:104493.

Ref 5:Liu M, et al. A multi-model deep convolutional neural network for automatic hippocampus segmentation and classification in Alzheimer's disease.Neuroimage. 2020 Mar;208:116459.

6. PLOS authors have the option to publish the peer review history of their article (what does this mean?). If published, this will include your full peer review and any attached files.

Reviewer #1: No

Reviewer #2: No

---

## [Author Response · Author response to Decision Letter 0]

2 Oct 2021

Reviewer #1: The manuscript “Interpreting coronary artery disease GWAS results: A functional genomics approach assessing biological significance” has an overall goal of addressing an important question, namely, when GWAS identifies multiple variants in a given locus is it due to the underlying LD or because of multiple variants with independent/only partially related functional effects? It also seeks to determine if incorporating tissue-wide eQTLs and splicing QTLs (sQTLs) is a “better” way to determine the underlying gene for a given non-coding locus. All of these efforts as a concept are important as we think about the non-coding genome and primarily non-coding variants identified from GWAS.

The strengths of the study are the overall conceptual framework posed (with caveat above that the analyses presented only take us so far); the careful incorporation of GTEx data coupled with individual level data obtained from the CATHGEN study through dbGaP; and the careful thought put into the genomic interpretations.

But while the overall conceptual framework and potential hypotheses are important, the manuscript’s impact is weakened due to the overall descriptive data presented; the expansive results section for primarily relatively straightforward bioinformatics work that would be done when evaluating any genetic variant; and the lack of functional validation of the proposed model outside of eQTLs to show that this approach is “better” than the comparator approach (which in this paper is just comparison to annotation in the Nikolay et al paper). In fact, for many of their SNPs, a simple search in dbSNP/UCSC confirms the gene that the SNP is in (not sure why the Nikolay paper annotated it differently). As such, the comparator being the Nikolay paper doesn’t help us determine if this method is “better”.

Thank you to the reviewer for so nicely highlighting the strengths and weaknesses of this work. We would agree that it is odd that despite the simplicity of much of what we have done and the thought that goes into many of these GWAS studies, tables of GWAS variants still report the nearest (often protein coding) gene. We have found that this is not unique to CAD or to the Nikpay paper and have presented an additional GWAS study as outlined below and in the revised manuscript to demonstrate this. We deleted the word ‘better’ as this value judgement is not needed. 

More minor issues include:

1) To address the above issue of the comparator being “GWAS annotated” but from a single paper, the authors should consider other phenotypes to perform these analyses;

Thank you for the suggestion! We have done a similar analysis to that outlined in Figure 1A for an additional GWAS study with variants reported by Lotta et al in Nature Genetics in 2017. This manuscript explored associations between genetic variants and insulin resistance phenotypes including higher fasting insulin levels adjusted for BMI, lower HDL cholesterol levels, and higher triglyceride levels, identifying 53 loci of interest. 

We find a similar pattern to that observed in the Nikpay paper in that many more candidate genes are introduced by considering expression and splicing analysis as well as location. 

2) The results section is very long and could be summarized more succinctly;

Agreed. We took out a significant portion of text and hope it now reads more easily.

3) Throughout, there needs to be more formal statistics done (and more statistics methods)…for example, 

Thank you for bringing this to our attention and highlighting some specific examples where we can help to bring more formal statistics and clarity. I think these changes substantially improve the manuscript and appreciate the comments. Please see the individual responses below.

in Table 2, ANOVA does not make sense for the model for association between SNPs and MI; 

We used an ANOVA to test nested generalized linear models. These GLMs incorporate different combinations of SNPs and were designed with the response variable of LIPA gene expression and then separately CAD. The p-value reported reflects the likelihood ratio test comparing the GLMs. In addition to the p-value we have added AICs as a measure of goodness of fit for each individual GLM.

what are p-values/effect sizes/directionality for eQTLs, 

These are p-values, effect sizes, and directionality that are reported by GTEx to reflect the association between the variant and gene expression. Details of their analysis can be found here https://www.gtexportal.org/home/documentationPage. Briefly, p-values reflect the alternative hypothesis that the slope in a linear regression model for normalized gene expression explained by a given genetic variant is non-zero. Effect size is this ‘non-zero’ slope and directionality corresponds to whether the minor allele is associated with higher or lower gene expression. We have added many of these details to the methods section to try and clarify further.

for “heat map” for clustering and showing that some variants have eQTLs in multiple tissues, etc.

Thank you for bringing this to our attention. The details of how this plot was generated had been inadvertently left out of the methods section. It is now included. We used the heatmap.2 function within the gplots R package to plot R2 for GTEx reported eQTLs. The colored bars above the heatmap were generated using the ColSideColors argument within the heatmap.2 function and are shaded to reflect the p-values with each color representing a different tissue type. 

4) For the LIPA analyses, why use MI when the original variants are CAD variants?

Thank you to the reviewer for highlighting this! We have tried to match definitions of CAD as best as possible across the datasets (see specific definitions below and track changes in the methods section of the text). Although these definitions had been used in the analysis, we were not diligent with the terminology in the initial submission and incorrectly used MI and CAD interchangeably.

CAD was defined in Nikpay as 

“Case status was defined by an inclusive CAD diagnosis (e.g. myocardial infarction (MI), acute coronary syndrome, chronic stable angina, or coronary stenosis >50%)”

CAD in CATHGEN

Non-zero CAD Index i.e. no CAD >50% stenosis (CADINDEX) or history of myocardial infarction (HHXMI) (https://ftp.ncbi.nlm.nih.gov/dbgap/studies/phs000703/phs000703.v1.p1/pheno_variable_summaries/phs000703.v1.pht003672.v1.CATHGEN_Metabolic_CVD_Risk_Subject_Phenotypes.data_dict.xml). 

CAD in GTEx

Recorded history of heart disease (MHHRTDIS) or heart attack (MHHRTATT) https://ftp.ncbi.nlm.nih.gov/dbgap/studies/phs000424/phs000424.v8.p2/pheno_variable_summaries/phs000424.v8.pht002742.v8.GTEx_Subject_Phenotypes.data_dict.xml).

5) The LIPA expression models are interesting and statistically a good way to further validate that multiple SNPs in a locus have independent effects and should be done across all the loci.

We are happy to see that our approach for validating multiple SNPs within a locus resonated with the reviewer and agree that it would be ideal to perform such an analysis across all loci. However for this to be done, candidate SNPs would need to be selected by hand for each locus, as was done with the LIPA locus where LD patterns of eQTLs (Figure 3) pointed to specific SNPs of interest. Currently this approach is too time consuming to be scaled up to the entire gene list. An alternative would be to scan large numbers of variants automatically, but then the burden of multiple hypothesis testing would likely obscure any true results. We look forward to continuing to address the possibility of multiple functional variants within a locus in future work.

6) For the LIPA expression models, need more formal statistics to show that adding SNPs improves models for expression (i.e. AIC, BIC, etc.)

Certainly. Thanks very much for the comment. We have included AIC values for each individual model in Table 2.

7) Overall, the manuscript could use an eye towards editorial improvement (uses colloquial language in several places like calling them “GWAS hits”, there are grammatical errors, results section is densely presented; catheterization in methods is spelled incorrectly; Wilcoxon is spelled “wilcoxin”; the test in R is wilcox.test not “wilcoxin.test”, etc.)

Thank you for bringing these to our attention. We have corrected the spelling errors you noted and more carefully reviewed the manuscript for additional grammatical and spelling errors.

8) How do the authors interpret the paradoxical results for rs1412444 on LIPA expression and MI?

Thanks for your question, it is one we have spent some time discussing. Complete absence of LIPA results in a rare genetic disorder known as Wolman disease in which lipids are not broken down in lysosomes, the liver upregulates cholesterol production, and early onset CAD as well as fatty deposition within various organs ensues. With this in mind, the association between decreased LIPA levels and MI is congruent. The question then remains why homozygous minor is associated with decreased rather than increased expression in this subset of individuals. It seems to imply the existence of a third, unknown factor regulating this relationship. For illustrative purposes, one potential explanation could be an additional variant that alters RNA transcript stability to be less stable and thus despite being produced in larger quantities is overall decreased.

9) How were eQTLs in GTEx defined? (again gets back to inclusion of more formal statistics). It’s hard to determine if some of the differences between variants/across tissues could just be variations around statistical significance without these more granular results.

Without a doubt variations around statistical significance are playing a role here! In initially preparing this manuscript, a previous version of GTEx with smaller sample size was used and there were significantly fewer GWAS variants that were also deemed eQTLs. As sample sizes increased, more eQTLs were detected and thus more overlap identified. In addition, tissue expression of any given gene can vary drastically, reducing the power to detect eQTLs at low expression levels measured with RNAseq. Therefore, interpreting eQTL differences between tissues must reflect RNA expression levels. 

To give readers more of a sense of how sample size may affect tissue specific differences in eQTLs we have included the sample size for each tissue alongside the tissue name in Figure 5. The concept that eQTLs depend on sample size (and tissue specific expression) remains within the text of the results section along with a reference to a supplementary figure that shows how power calculations for a hypothetical SNP with minor allele frequency 0.05 and an effect size of 40% (i.e. no minor alleles is 20% less than the median tissue specific gene expression and two minor alleles is 20% greater than the median tissue specific gene expression) compare across tissues.

 

Reviewer #2: In the manuscript entitled “Interpreting coronary artery disease GWAS results: A functional genomics approach assessing biological significance”, the authors explored the complexity of each of CAD loci by use of tissue specific RNA sequencing data from GTEx to identify genes that exhibit altered expression patterns in the context of GWAS significant loci,and expanded the list of candidate genes from the 75 currently annotated by GWAS to 245.

The following papers can be cited and followed for the meta-analytic procedures (if the data is not enough available, at least DISCUSSION should be added as the LIMITATION of this study with enough citation to support the viewpoints):

Ref 1: Wu Y, et al. Multi-trait analysis for genome-wide association study of five psychiatric disorders. Transl Psychiatry. 2020 Jun 30;10(1):209.

Ref 2:Jiang L, et al. Sex-Specific Association of Circulating Ferritin Level and Risk of Type 2 Diabetes: A Dose-Response Meta-Analysis of Prospective Studies..J Clin Endocrinol Metab. 2019 Oct 1;104(10):4539-4551.

Ref 3: Xu M, et al. Quantitative assessment of the effect of angiotensinogen gene polymorphisms on the risk of coronary heart disease. Circulation. 2007 Sep 18;116(12):1356-66

Trans-ethnitic and trans-trait meta-analysis of cardiometabolic traitscan be referred to Ref 1.

Subgroup analyses based on sex, age, race, gene dosage can be referred to Ref 2. and3

We thank the reviewer for bringing these references to our attention. However as we have not undertaken a meta-analysis in this manuscript, instead working within several large scale databases (GTEx and CATHGEN) without combining them, we did not find these references would be relevant. We have added text to the methods section to help to clarify this point and reserve these references for any future meta-analytic work to which they may be more applicable.

Integrating GWAS signals with eQTL from GTEX or pQTLs is a good strategy to exploring the causality of the genetic varients in the development of cardiometabolic traits. But I strong suggest to do causal inference analysis to see if the GWAS signals are causally triggering the develop,ent of CAD through mediating the expression of given genes in specific tissues.

We thank the reviewer for their comment. Indeed causal analysis is a critical piece as GWAS findings are essentially associations and nothing more. We have added additional text to the discussion emphasizing the need for molecular biology experimentation to establish causal roles for genetic variation. 

In addition, the significantly associated SNPs may be used to predict disease susceptibility in the context of its influence of gene expression,therefore, the authors may explore the possibility to conduct a machine-learning model to predict CAD risk or cardiometalobic traits based these significant SNPs. For this reason, the authors may cite the following papers to follow these references’ procedure to construct a standard prediction model based on the significant SNPs (probably include the gene expression information). Deep learning is a hot topic in dissecting the genome variants’ roles in the phenome. Especially deep learning method is a very promising way to predict disease risk based on clinical information and genetic biomarkers(If deep learning can not be used, please discuss as the LIMITATION of this study with enough citation to support the viewpoints).

Ref 4:Yu H, et al. LEPR hypomethylation is significantly associated with gastric cancer in males.Exp Mol Pathol. 2020 Oct;116:104493.

Ref 5:Liu M, et al. A multi-model deep convolutional neural network for automatic hippocampus segmentation and classification in Alzheimer's disease.Neuroimage. 2020 Mar;208:116459.

Thanks to the reviewer for the provided references and for the suggestion. We would agree that machine learning would be an interesting and potentially very powerful methodology to account for disease risk with various data points including gene expression and genetic variation. To a large extent, we are limited by a combination of sample size and data availability. So these analyses were not feasible. We have added text to the discussion emphasizing the need for future directions to explore machine learning methodology and to incorporate other markers of genetic variation beyond SNPs such as methylation and referenced the recommended publications.

---

## [Editor Report · Decision Letter 1]

3 Jan 2022

Interpreting coronary artery disease GWAS results: A functional genomics approach assessing biological significance

PONE-D-20-39555R1

Dear Dr. Katherine Hartmann,

We’re pleased to inform you that your manuscript has been judged scientifically suitable for publication and will be formally accepted for publication once it meets all outstanding technical requirements.

Kind regards,

Mingqing Xu

Academic Editor

PLOS ONE

Additional Editor Comments (optional):

It can be accepted for publication now.
---

## [Editor Report · Acceptance letter]

7 Feb 2022

PONE-D-20-39555R1 

Interpreting coronary artery disease GWAS results: A functional genomics approach assessing biological significance 

Dear Dr. Hartmann:

I'm pleased to inform you that your manuscript has been deemed suitable for publication in PLOS ONE. Congratulations! Your manuscript is now with our production department. 

Kind regards, 

on behalf of

Dr. Mingqing Xu 

Academic Editor

PLOS ONE